# The Association between *ABCG2* 421C>A (rs2231142) Polymorphism and Rosuvastatin Pharmacokinetics: A Systematic Review and Meta-Analysis

**DOI:** 10.3390/pharmaceutics14030501

**Published:** 2022-02-24

**Authors:** Yubin Song, Hee-Hyun Lim, Jeong Yee, Ha-Young Yoon, Hye-Sun Gwak

**Affiliations:** College of Pharmacy and Graduate School of Pharmaceutical Sciences, Ewha Womans University, Seoul 03760, Korea; 1564069@ewhain.net (Y.S.); dlagmlgus000@ewhain.net (H.-H.L.); jjjhello1@naver.com (J.Y.); hayoungdymphnayoon@gmail.com (H.-Y.Y.)

**Keywords:** *ABCG2* 421C>A, rosuvastatin, meta-analysis, pharmacokinetics, polymorphism, systematic review

## Abstract

Although several studies have revealed the association between rosuvastatin pharmacokinetics and the *ABCG2* 421C>A (rs2231142) polymorphism, most studies were conducted with small sample sizes, making it challenging to apply the findings clinically. Therefore, the purpose of this study is to perform a meta-analysis of the relationship between the *ABCG2* 421C>A polymorphism and rosuvastatin pharmacokinetics. We searched three electronic databases, EMBASE, PubMed, and Web of Science, using search terms related to *ABCG2* gene polymorphisms and rosuvastatin. In addition, we reviewed studies published before 12 August 2021, to examine the relationship between the *ABCG2* 421C>A polymorphism and rosuvastatin pharmacokinetics. To examine the magnitude of the association, the log geometric mean difference (lnGM) and 95% confidence intervals (CIs) were calculated and interpreted as the antilogarithm of a natural logarithm (e^lnGM^). The meta-analysis was performed using Review Manager (version 5.4) and R Studio (version 4.0.2). Subgroup analysis was performed according to race and the types of mean values. Among the 318 identified studies, a total of 8 studies involving 423 patients is included in this meta-analysis. The A allele carriers of *ABCG2* 421C>A showed 1.5 times higher in both AUC_0-∞_ (lnGM = 0.43; 95% CI = 0.35–0.50; *p* < 0.00001) and C_max_ (lnGM = 0.42; 95% CI = 0.33–0.51; *p* < 0.00001) than non-carriers, while there was no significant difference in T_max_ and half-life. There was no significance in the pharmacokinetic parameters of the subgroups using either ethnicity or mean values. This meta-analysis demonstrates that subjects carrying the A allele of *ABCG2* 421C>A show significantly increased AUC_0-∞_ and C_max_ values compared to subjects with the CC genotype. Therefore, information about *ABCG2* genotypes might be useful for individualized rosuvastatin therapy.

## 1. Introduction

Rosuvastatin, a potent 3-hydroxy-3-methylglutaryl-CoA (HMG CoA) reductase inhibitor [1], markedly reduces serum levels of low-density lipoprotein (LDL) cholesterol and is used for the treatment of dyslipidemia and the prevention of coronary heart disease [2]. The major disposition pathways of rosuvastatin are predominantly modulated by the expression and activity of several intestinal and hepatobiliary membrane transporters [3]. In enterocytes, intestinal influx carriers, such as organic anion transporting polypeptide (OATP) family members 1A2, 1B1, and 2B1, regulate rosuvastatin disposition, while breast cancer resistance protein (BCRP) and multidrug resistance-associated protein 2 (MRP2) mediate efflux transport [4]. In hepatocytes, the sodium taurocholate co-transporting polypeptide (NTCP), OATP1B1, OATP1B3, and OATP2B1 transport rosuvastatin into the cytoplasm, while BCRP and MRP2 excrete rosuvastatin into bile [5].

It has been reported that interindividual variation in exposure to rosuvastatin is associated with the polymorphisms of OATP1B1 and BCRP, which are encoded by the solute carrier organic anion transporter family member 1B1 (*SLCO1B1)* and the ATP-binding cassette superfamily G member 2 (*ABCG2)*, respectively. [4]. The association between *SLCO1B1* and rosuvastatin pharmacokinetics has been studied extensively, while little is known about the relationship between *ABCG2* and rosuvastatin pharmacokinetics. 

ABCG2 is generally expressed in tissues, such as the brain, kidney, liver, placenta, and small intestine [6]. It acts as a tissue barrier by not only limiting the absorption of its substrates from the gastrointestinal tract, blood–brain barrier, and placenta, but also increasing the excretion of its substrates into the bile and urine [7]. Several xenobiotics, such as antibiotics, cytotoxic agents, and anticancer drugs, as well as endogenous compounds—estrogens and estrogen conjugates—are substrates of *ABCG2* [8]. 

The *ABCG2* 421C>A variant (Gln141Lys; rs2231142), one of the most frequent polymorphisms in the *ABCG2* gene [8], decreases in vitro *ABCG2* efflux activity [9]. As rosuvastatin is an *ABCG2* substrate, the *ABCG2* 421C>A polymorphism can affect rosuvastatin pharmacokinetics and its therapeutic efficacy. Although several studies reported the association between rosuvastatin pharmacokinetics and *ABCG2* 421C>A polymorphism [10], most studies were conducted with a small number of patients, resulting in large variability that makes it difficult to apply the findings clinically. Therefore, the goal of this meta-analysis was to combine the available data from studies to obtain more robust pharmacokinetic estimates and demonstrate the potential roles of the *ABCG2* 421C>A polymorphism in rosuvastatin pharmacokinetics that may affect its efficacy and side effects.

## 2. Materials and Methods

### 2.1. Search Strategy and Study Selection

This meta-analysis was performed based on the Preferred Reporting Items for Systematic Reviews and Meta-Analyses (PRISMA) checklist [11]. Two researchers independently searched for studies published before 12 August 2021. An extensive search of electronic databases (PubMed, Web of Science, and EMBASE) was performed with the following search terms: (Rosuvastatin OR (Rosuvastatin Calcium) OR Crestor OR (ZD 4522) OR ZD4522) AND (*ABCG2* OR (ATP-Binding Cassette Transporter, Subfamily G, Member 2) OR (ATP-Binding Cassette Superfamily G member 2) OR BCRP OR BCRP1 OR (Breast cancer resistance protein) OR MRX OR MXR OR MXR1 OR MXR-1 OR CD338 OR CDw338) AND (polymorph* OR variant* OR mutation* OR genotyp* OR phenotyp* OR haplotyp* OR allele* OR SNP*). 

After removing duplicates, two researchers independently screened the titles and abstracts of all records to identify potentially eligible studies. Then, a full-text review was performed to determine the final inclusion according to the eligibility criteria. In cases of disagreement, a consensus was reached by discussion.

Studies were included if they (i) evaluated the association of the *ABCG2* 421C>A polymorphism with rosuvastatin pharmacokinetic parameters, and (ii) included healthy adult volunteers receiving a single dose of rosuvastatin. Studies were excluded if they were (i) non-original articles, (ii) performed on subjects receiving concomitant drugs, (iii) unable to extract data, (iv) overlapping papers, and (v) not written in English. Only the most recent and comprehensive data were included in the meta-analysis in the case of overlapping data.

### 2.2. Data Extraction and Study Quality Assessment

Two researchers independently extracted data, and the discrepancies were resolved by consensus. The extracted data included the following information: the name of the first author, publication year, nation, race, number of patients, percentage of males, age, body mass index, rosuvastatin dose, studied pharmacokinetic parameters, type of mean values, and genotyping and quantitative methods. In addition, the pharmacokinetic parameters of rosuvastatin impacted by *ABCG2* 421C>A were also extracted from the studies. The primary outcomes were the area under the plasma concentration–time curve from 0 h to infinity (AUC*_0-∞_*) and the peak plasma concentration (C_max_); the secondary outcomes were the time to reach C_max_ (T_max_) and the half-life.

Two researchers assessed articles based on the Newcastle–Ottawa Scale (NOS) [12], evaluating studies in the three categories of selection (0–4 points), comparability (0–2 points), and outcome assessment (0–3 points). The score range of NOS is from 0 to 9.

### 2.3. Statistical Analysis

The meta-analysis was conducted using Review Manager (version 5.4; The Cochrane Collaboration, Copenhagen, Denmark), using inverse variance weighting. The log geometric mean difference (lnGM) and their corresponding 95% confidential intervals (95% CIs) were used to identify the relationship of the *ABCG2* 421C>A polymorphism with rosuvastatin AUC_0-∞_ and C_max_. The arithmetic mean difference (AMD) and their corresponding 95% CIs were used for determining the T_max_ and half-life. If the administered rosuvastatin dose was different from 20 mg, the mean and standard deviation (SD) were adjusted, given that AUC_0-∞_ and C_max_ were reported to increase proportionally with the dose [13]. To pool the arithmetic mean (AM) and geometric mean (GM), we calculated GM and geometric standard deviations (GSDs) to perform the meta-analysis using the conversion equations of Higgins et al. [14].
$$ln\left(GM\right)=ln\;\left(AM\right)-0.5ln\left(1+\frac{SD^{2}}{AM^{2}}\right)$$
$$ln\;\left(GSD\right)=\sqrt{ln(1+\frac{SD^{2}}{AM^{2}}})$$

The magnitude of the association was interpreted as the antilogarithm of a natural logarithm (e^lnGM^). The heterogeneity across studies was estimated using a chi-square test and an *I^2^* statistic [15]. A random-effects model was applied when heterogeneity existed (*I^2^* > 50%); otherwise, the fixed-effects model was applied [16]. Both Begg’s rank correlation test and Egger’s regression test of the funnel plot were performed using R Studio software (version 4.0.2; R Foundation for Statistical Computing, Vienna, Austria) to detect publication bias [17,18]. Sensitivity analysis was conducted by the sequential omission of each study to validate the robustness of the results. Subgroup analysis was performed on the basis of ethnicity and the reported type of mean values, and the chi-square test was used for subgroup comparisons. A *p*-value < 0.05 was considered statistically significant.

## 3. Results

A detailed flow chart of the study selection process is presented in Figure 1. A total of 318 studies was identified from the searches of 3 databases. After 113 duplicates were excluded, 205 records were initially identified, of which the titles and abstracts were screened for eligibility for inclusion in this study. Thirty-six studies were selected for full-text reviews and assessed for eligibility from this initial review. Among these 36 studies, 28 studies were excluded for the following reasons: non-original articles (*n* = 5), an in vitro or animal study (*n* = 1), studies performed on subjects receiving concomitant drugs (*n* = 5), studies without pharmacokinetic parameters (*n* = 3), studies on other genes (*n* = 3), a study on other drugs (*n* = 1), studies unable to extract data (*n* = 9), and studies not written in English (*n* = 1). Finally, a total of eight studies were selected for meta-analysis [3,8,10,19,20,21,22,23].

The characteristics of the included studies are presented in Table 1. The studies were published from 2006 to 2020. Three of them were conducted in China, two in the U.S.A., one in Canada, one in Finland, and one in Korea. Six of them were cohort studies, and two were randomized controlled trials. For the primary outcomes, four studies used GM, whereas four studies used AM. NOS scores ranged from 6 to 7.

Eight studies involving 423 patients were evaluated to investigate the association between the *ABCG2* 421C>A polymorphism and rosuvastatin pharmacokinetic parameters (Appendix A). A meta-analysis of the pooled data showed a significant association between the *ABCG2* 421C>A polymorphism and rosuvastatin AUC_0-∞_ and C_max_ values. The A allele carriers of *ABCG2* 421C>A showed 1.5 times higher in both AUC_0-∞_ (lnGM = 0.43; 95% CI = 0.35–0.50; *p* < 0.00001, Figure 2a) and C_max_ (lnGM = 0.42; 95% CI = 0.33–0.51; *p* < 0.00001, Figure 2b) than non-carriers. Neither Begg’s test nor Egger’s test showed significant publication bias for these studies (all *p* > 0.05, Appendix A). There was no significant difference in T_max_ (Figure 2c) and half-life (Figure 2d). The tests for detecting publication bias showed that statistically significant publication bias existed for half-life (Begg’s test, z = 2.04, *p* = 0.0415; Egger’s test, t = 4.57, *p* = 0.0446) (Appendix A), but not for T_max_ (*p* > 0.05) (Appendix A).

We performed a sensitivity analysis by sequentially omitting each study. The results for lnGM for AUC_0-∞_ (lnGM range 0.42–0.45) and C_max_ (lnGM range 0.39–0.44) and AMD for T_max_ (AMD range −0.12–0.15), and half-life (AMD range −0.57–0.21) were similar to the main results (Appendix A). 

The *ABCG2* 421C>A polymorphism showed a similar trend in the subgroup analysis of AUC_0-∞_ and C_max_ of rosuvastatin for Caucasian and Asian populations (Figure 3). However, there was no significant difference between the two subgroups (*p* = 0.61 and *p* = 0.54, respectively).

Moreover, we found no significant differences on the AUC_0-∞_ and C_max_ of rosuvastatin between GM and AM (*p =* 0.63 and *p =* 0.67, respectively; Figure 4) in another subgroup analysis conducted for studies with these two different types of mean values.

## 4. Discussion

To our knowledge, this is the first meta-analysis to reveal an association between rosuvastatin pharmacokinetics and the *ABCG2* 421C>A polymorphism. Both AUC_0-∞_ and C_max_ were increased by 50% in A allele carriers compared to CC genotype subjects. However, there were no significant differences in T_max_ and half-life between *ABCG2* 421 CC and CA/AA genotype subjects.

As the studies by Huguet et al. [3] and Kim et al. [21] had a small sample size and high variance, the corresponding study weights in the meta-analysis were approximately zero. The study by Huguet et al. was not included in the analysis of C_max_ as its log-transformed standardized deviation value was negative [3]. As Liu et al. collected and analyzed blood samples for up to 96 h after rosuvastatin ingestion, we considered the AUC_0-96_ value as the AUC_0-∞_ value [22]. 

In addition to AUC_0-∞_ and C_max_, several studies showed that apparent oral clearance (Cl/F) was also associated with *ABCG2* 421 C>A. According to Zhang et al. [10], CL/F (l/h) was higher in 421 A allele carriers than in CC genotype carriers (674.0 vs. 384.7). Liu et al. [22] and Wan et al. [23] showed similar results. However, in the study of Keskitalo et al., renal clearance was not statistically significant among 421 C>A genotypes, although the rosuvastatin amount excreted in urine within 24 h was higher in AA genotype carriers than in C allele carriers.

Our results with healthy volunteers were similar to previous results with hypercholesterolemia patients. Lee et al. revealed that in Chinese patients with hypercholesteremia, the mean plasma concentrations of rosuvastatin and its metabolite were higher in subjects with the *ABCG2* 421 AA genotype than in those with the CA genotype or CC genotype [24]. In addition, DeGorter et al., who used multiple linear regression analysis, reported that 421 A allele carriers had higher plasma rosuvastatin concentrations (*p <* 0.05) [25]. 

Several studies reported that 421 C>A affected not only pharmacokinetic properties, but also drug responses of rosuvastatin. According to Lee et al., % changes in LDL-C were significantly different among 421 C>A genotypes in rosuvastatin-treated patients (CC vs. CA vs. AA: −48.5 vs. −55.1 vs. −54.9, *p* <10^−5^) [24]. Similarly, Tomlinson et al. and Kozhakhmetov et al. showed that 421 A allele carriers had greater reductions in LCL-C and total cholesterol in patients treated with rosuvastatin [26,27]. In terms of adverse effects, Merćep et al. reported that 421 C>A increased two times higher risks of rosuvastatin-related myotoxicity and hepatotoxicity [28]. 

Recently, the Clinical Pharmacogenetics Implementation Consortium (CPIC) conducted a systematic review and published the guideline for statin-associated with musculoskeletal symptoms, including *ABCG2* 421 C>A [29]. According to the CPIC guideline, a rosuvastatin dose of ≤ 20 mg is recommended as the starting dose for individuals with *ABCG2* poor function due to the high exposure. In line with its recommendation, this study provided quantitative evidence for the associations between *ABCG2* 421 C>A and increased AUC_0-∞_ and C_max_ of rosuvastatin by pooling the results of related articles identified by systematic review through the meta-analysis.

It has been reported that the *ABCG2* 421C>A polymorphism decreases the plasma membrane expression or transporter ATPase activity of *ABCG2* [9,30,31,32]. Reduced activity of ABCG2 in 421 A allele carriers increases the absorption of rosuvastatin in the gastrointestinal tract while decreasing drug efflux in biliary ducts. Therefore, drug accumulation in the systemic circulation is caused by the dual effects of enhanced absorption and reduced hepatic clearance [10].

Several meta-analyses have shown the association between the *ABCG2* 421C>A polymorphism and drug responses. For example, a meta-analysis of the association between *ABCG2* 421C>A polymorphism and imatinib response in patients with chronic myeloid leukemia revealed that the A allele was significantly associated with an increased rate of overall response [33]. Similar results were found in another meta-analysis of Kurdish breast cancer patients; patients with the C allele showed poorer responses to anthracyclines and paclitaxel treatments than those with the AA genotype [34]. Reduced activity of *ABCG2* in A allele carriers appears to be the reason for this low efficacy.

Subgroup analysis by ethnicity showed that, regardless of ethnicity, 421 A allele increased the rosuvastatin exposure by 50%. As the allele frequency of 421 A was higher in Asians than Caucasians (0.29 vs. 0.15) [35], Asians may be more affected by this allele than Caucasians, thereby having increased exposure to rosuvastatin. In line with our results, the absolute oral bioavailability of rosuvastatin is higher in Asians than in Caucasians (29% vs. 20%) [36]. The Food and Drug Administration (FDA) recommended reducing the rosuvastatin dose in Asian patients, due to the increased toxicity of rosuvastatin in Asian patients [37]. 

Another subgroup analysis by reported type of mean values showed that the GM method and AM method showed similar results, supporting the idea that the Higgins conversion equation [14] could provide reasonable estimates for pooling the results in the meta-analysis. Therefore, this equation can be an appropriate solution to include studies with both types of mean values in meta-analysis and derive strong evidence. 

This study has some limitations. First, we conducted a meta-analysis on healthy volunteers, not patients, which may result in low external validity. Second, we could not consider other intrinsic or extrinsic factors that could affect rosuvastatin pharmacokinetics. Third, the number of studies for T_max_ and half-life was relatively small. Despite those limitations, this meta-analysis revealed that A allele carriers showed increased AUC_0-∞_ and C_max_ values compared to subjects with the CC genotype. Thus, as there might be interindividual variations in rosuvastatin efficacy and toxicity, information about *ABCG2* genotypes might be useful for individualized rosuvastatin therapy.

## Figures and Tables

**Figure 1 pharmaceutics-14-00501-f001:**
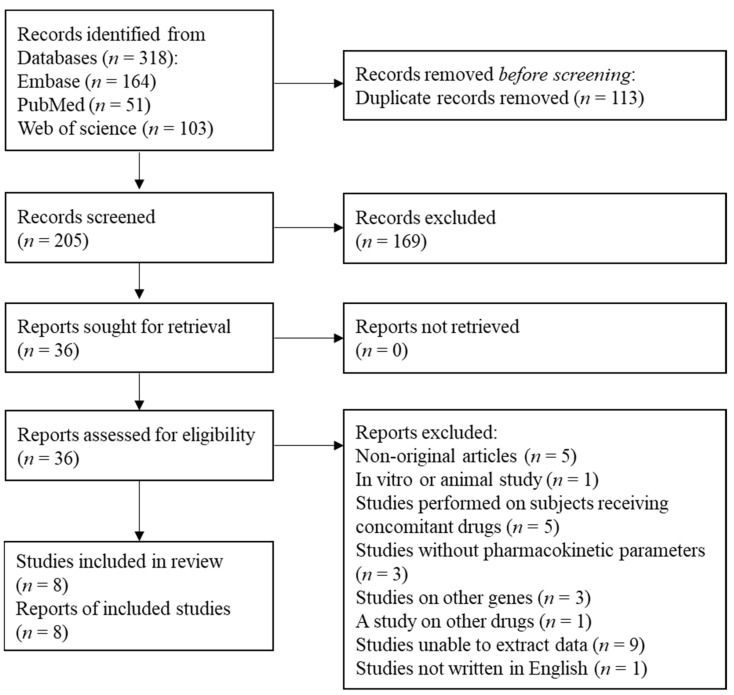
Preferred reporting items for systematic reviews and meta-analyses (PRISMA) flow diagram of the study selection.

**Figure 2 pharmaceutics-14-00501-f002:**
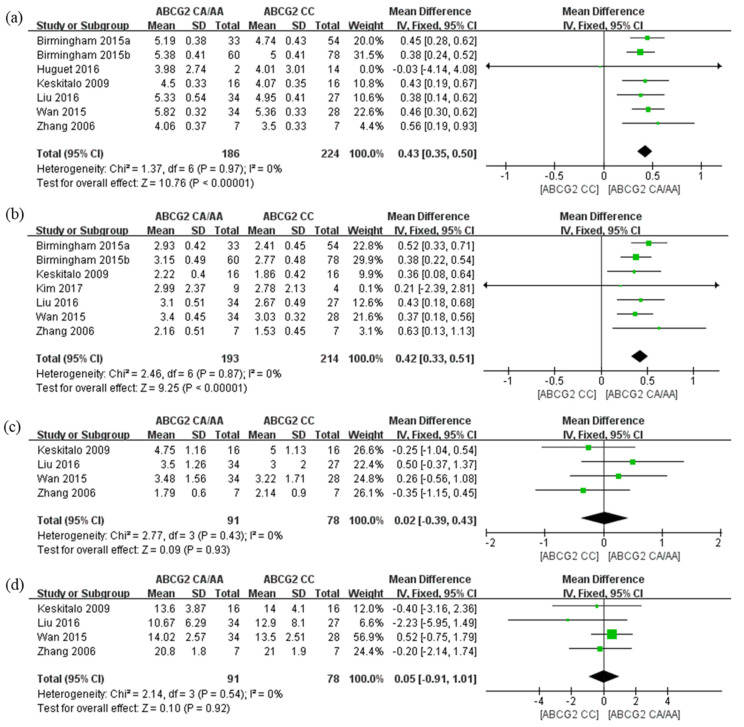
Forest plots of the effects of *ABCG2* 421C>A on rosuvastatin; values expressed as logarithm of geometric mean for (**a**) AUC*_0-∞_* and (**b**) C_max_; arithmetic mean differences for (**c**) T_max_ and (**d**) half-life. The effect size for each individual study is represented by the green square and the overall pooled effect size is represented by the black diamond.

**Figure 3 pharmaceutics-14-00501-f003:**
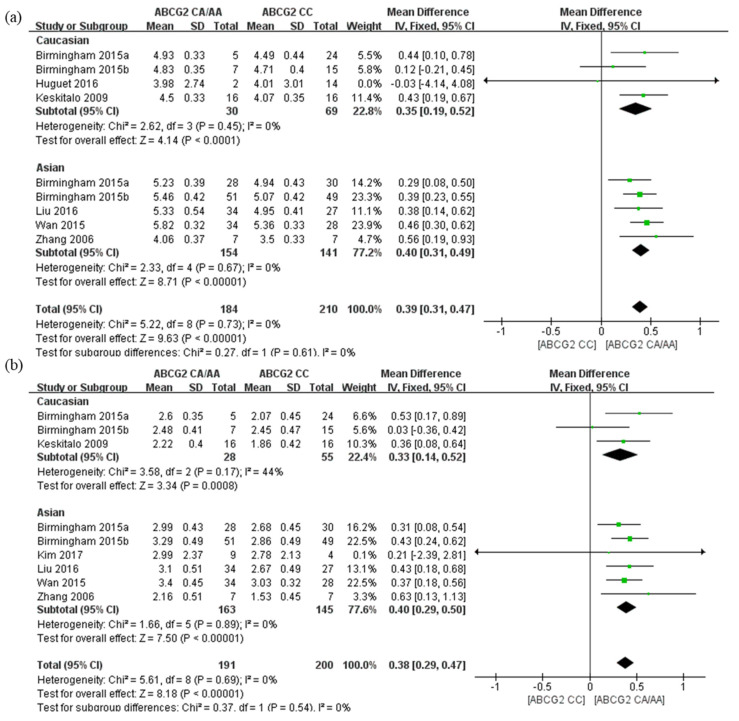
Forest plots of the effects of *ABCG2* 421C>A on rosuvastatin; values expressed as logarithm of geometric mean (**a**) AUC*_0-∞_* and (**b**) C_max_ in healthy Caucasian and healthy Asian subjects. The effect size for each individual study is represented by the green square and the overall pooled effect size is represented by the black diamond.

**Figure 4 pharmaceutics-14-00501-f004:**
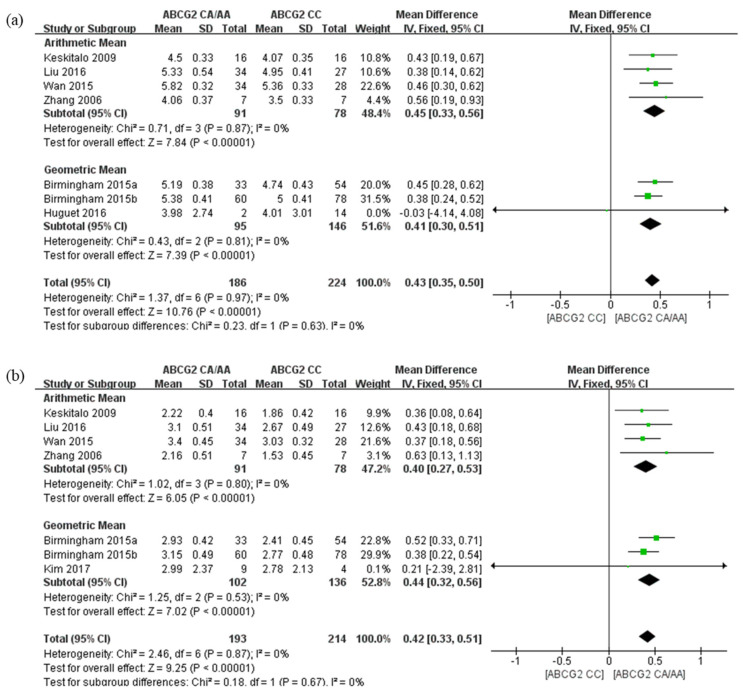
Forest plots of the effects of *ABCG2* 421C>A on rosuvastatin; values expressed as logarithm of geometric mean (**a**) AUC*_0-∞_* and (**b**) C_max_ with measured values of studies given as geometric means and arithmetic means. The effect size for each individual study is represented by the green square and the overall pooled effect size is represented by the black diamond.

**Table 1 pharmaceutics-14-00501-t001:** Characteristics of the studies included.

First Author, Year	Nation	Race	N (Male Percent)	Age, Year (SD)	BMI, kg/m^2^ (SD)	Rosuvastatin Dose (mg)	PK Outcomes	Type of Mean Value	Genotyping Methods	Quantitative Methods	NOS
Birmingham et al., 2015 [19]	U.S.A.	Caucasian, Chinese, Japanese	93 (65.6)	36.2 (14.3)	23.8 (2.9)	20	AUC, C_max_	Geometric	TaqMan assay	LC-MS/MS	7
Birmingham et al., 2015 [20]	U.S.A.	Asian-Indian, Caucasian, Pooled-Asian ^a^	184 (66.3)	23.0 (10.8)	23.7 (3.0)	20	AUC, C_max_	Geometric	TaqMan assay	HPLC-MS/MS	7
Huguet et al., 2016 [3]	Canada	Caucasian	16 (100.0)	28.0 (8.0)	24.1 (2.2)	10	AUC	Geometric	TaqMan assay	HPLC-HESI-MS/MS	7
Keskitalo et al., 2009 [8]	Finland	Caucasian	32 (50.0)	22.3 (2.6)	N/A	20	AUC, C_max_, t_max_ ^c^, half-life	Arithmetic	TaqMan assay	LC-MS/MS	7
Kim et al., 2017 [21]	Korea	Korean	13 (15.4)	26.8 (4.0)	N/A	20	C_max_	Geometric	Pyrosequencing assay	LC-MS/MS	6
Liu et al., 2016 [22]	China	Chinese	61 (100.0)	20–32 ^b^	18–24 ^b^	20	AUC, C_max_, t_max_, half-life	Arithmetic	MALDI-TOF	LC-MS/MS	7
Wan et al., 2015 [23]	China	Chinese	62 (100.0)	18–24 ^b^	18–24 ^b^	10	AUC, C_max_, t_max_, half-life	Arithmetic	Pyrosequencing assay	HPLC-MS/MS	6
Zhang et al., 2006 [10]	China	Chinese	14 (100.0)	N/A	N/A	20	AUC, C_max_, t_max_, half-life	Arithmetic	Direct sequencing	LC-MS/MS	7

AUC: area under the curve; BMI: body mass index; HPLC-HESI-MS/MS: high performance liquid chromatography-heated electrospray ionization-tandem mass spectrometry; HPLC-MS/MS: high performance liquid chromatography with tandem mass spectrometry; LC-MS/MS: liquid chromatography with tandem mass spectrometry; MALDI-TOF: matrix-assisted laser desorption ionization-time of flight mass spectrometry; N/A: not available; NOS: Newcastle–Ottawa score; PK: pharmacokinetics; SD: standard deviation; U.S.A.: United States of America. ^a^ Chinese, Filipino, Japanese, Korean, Vietnamese; ^b^ range; ^c^ median and range.

## Data Availability

Not applicable.

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
