# Peer review of "The Association between ABCG2 421C>A (rs2231142) Polymorphism and Rosuvastatin Pharmacokinetics: A Systematic Review and Meta-Analysis"

_pharmaceutics, 2022, doi:10.3390/pharmaceutics14030501_

Round 1

Reviewer 1 Report

The authors performed meta-analysis considering the association between rosuvastatin pharmacokinetics and ABCG2 421C>A gene polymorphism. It was an interesting topic and I feel it will meet the same opinion by the readers. The core of the study was done correctly. Still, I have some minor comments and suggestions:

  1. In think that abstract should included the number of all the studies you have reviewed not only the final 8.
  2. You should expand discussion related to this polymorphism and clinical outcomes. It can be of particular importance since you included only studies with the healthy subjects. 
  3.  Figures should have better quality.
  4.  Is there a chance to compare some other PK parameters in relation to ABCG2 genotype? Explanation will be enough.

Author Response

The authors performed meta-analysis considering the association between rosuvastatin pharmacokinetics and ABCG2 421C>A gene polymorphism. It was an interesting topic and I feel it will meet the same opinion by the readers. The core of the study was done correctly. Still, I have some minor comments and suggestions:

  1. In think that abstract should included the number of all the studies you have reviewed not only the final 8.

-> As commented, we revised the sentences as follows: Among 318 studies identified, a total of eight studies involving 423 patients were included in this meta-analysis.

  1. You should expand discussion related to this polymorphism and clinical outcomes. It can be of particular importance since you included only studies with the healthy subjects.

-> As commented, we added the sentences in Discussion sections as follows: Several studies reported that 421 C>A affected not only pharmacokinetic properties but also drug responses of rosuvastatin. According to Lee et al., % changes in LDL-C were significantly different among 421 C>A genotypes in rosuvastatin-treated patients (CC vs CA vs AA: -48.5 vs -55.1 vs -54.9, p <10-5) [24]. Similarly, Tomlinson et al. and Kozhakhmetov et al. showed that 421 A allele carriers had greater reductions in LCL-C and total cholesterol in patients treated with rosuvastatin [26, 27]. In terms of adverse effects, Merćep et al. reported that 421 C>A increased two times higher risks of rosuvastatin-related myotoxicity and hepatotoxicity [28].

  1. Figures should have better quality.

-> As commented, we updated the figures.

  1. Is there a chance to compare some other PK parameters in relation to ABCG2 genotype? Explanation will be enough.

-> As commented, we added the sentences in Discussion as follows: In addition to AUC0-∞ and Cmax, several studies showed that apparent oral clearance (Cl/F) was also associated with ABCG2 421 C>A. According to Zhang et al. [10], CL/F (l/h) was higher in 421 A allele carriers than in CC genotype carriers (674.0 vs 384.7). Liu et al. [22] and Wan et al. [23] showed similar results. However, in the study of Keskitalo et al., renal clearance was not statistically significant among 421 C>A genotypes, although the amount of rosuvastatin excreted in urine within 24 hours was higher in AA genotype carriers than in C allele carriers.

Reviewer 2 Report

The reviewer recommends a revision on the manuscript considering but not being limited to followings:

  1. The meta-analysis included the journal articles that had been published until August 12, 2020. The reviewer recommends updating the manuscript if there are more recent publications.
  2. It is unclear how the authors weighted each study used in the analysis (e.g., Huguet et al. 0.0%). The reviewer recommends including the weighting method in the method section and discussing the impact of weighting in the discussion section.
  3. It is confusing whether “reviewers,” “investigators” and “researchers” are different or not. The reviewer recommends using a consistent term.
  4. The reviewer recommends including the actual values of Cmax and AUC of rosuvastatin reported in the primary articles that the authors used for meta-analysis.
  5. The exclusion criteria do not match in Figure 1 and Section 2.1. Search strategy and study selection section.
  6. Line 25, 151. The reviewer recommends adding “both” in the sentences to make sure that Cmax is also 1.5 times higher in ABCG2 421C>A allele carriers than non-carriers.
  7. Line 86. The reviewer recommends adding the term not “written” in English.
  8. Line 102, 104. It seems to be clear to add the term “their corresponding” 95% CI.
  9. Line 113. The reviewer recommends adding a reference for I2 statistics method.
  10. Line 197. The sentence starting from “According to…” appears to be out of place.
  11. Line 172. The reviewer recommends stating in the method section what statistical test was used to determine the significance (i.e., p=0.61).
  12. Line 200. The reviewer recommends adding references to be consistent with “Several studies”.
  13. Line 213-222. The reviewer recommends revising the whole paragraph. It is very difficult to comprehend.
  14. Line 223-225. The reviewer recommends revising the whole paragraph. It is very difficult to comprehend.

Author Response

The reviewer recommends a revision on the manuscript considering but not being limited to followings:

  1. The meta-analysis included the journal articles that had been published until August 12, 2020. The reviewer recommends updating the manuscript if there are more recent publications.

-> We searched the articles until August 12, 2021. The number ‘2020’ was a typo. In addition, we tried to search recent publications; although the number of identified studies was slightly increased (318 -> 342), there were no additional articles that met the inclusion criteria.

  1. It is unclear how the authors weighted each study used in the analysis (e.g., Huguet et al. 0.0%). The reviewer recommends including the weighting method in the method section and discussing the impact of weighting in the discussion section.

-> We added the phrases in the method section: … using inverse variance weighting.

-> We added the sentences in the discussion section: As the studies by Huguet et al. [3] and Kim et al. [21] had a small sample size and high variance, the corresponding study weights in the meta-analysis were approximately zero.

  1. It is confusing whether “reviewers,” “investigators” and “researchers” are different or not. The reviewer recommends using a consistent term.

-> As commented, we used the word ‘researchers’.

  1. The reviewer recommends including the actual values of Cmax and AUC of rosuvastatin reported in the primary articles that the authors used for meta-analysis.

-> We added the actual values of Cmax and AUC of rosuvastatin in Supplementary Table 1.

  1. The exclusion criteria do not match in Figure 1 and Section 2.1. Search strategy and study selection section.

-> As commented, we revised the words in Figure 1 and Results.

  1. Line 25, 151. The reviewer recommends adding “both” in the sentences to make sure that Cmax is also 1.5 times higher in ABCG2 421C>A allele carriers than non-carriers.

-> We added “both”.

  1. Line 86. The reviewer recommends adding the term not “written” in English.

-> We added the word “written”.

  1. Line 102, 104. It seems to be clear to add the term “their corresponding” 95% CI.

-> We added the words “their corresponding”.

  1. Line 113. The reviewer recommends adding a reference for I2 statistics method.

-> We added the reference for I2 statistics.

[15] Higgins JP, Thompson SG. Quantifying heterogeneity in a meta-analysis. Stat Med. 2002;21(11):1539-58. doi: 10.1002/sim.1186.

  1. Line 197. The sentence starting from “According to…” appears to be out of place.

-> As commented, we revised the sentences as follows: Similarly, Tomlinson et al. and Kozhakhmetov et al. showed that 421 A allele carriers had greater reductions in LCL-C and total cholesterol in patients treated with rosuvastatin [26, 27].

  1. Line 172. The reviewer recommends stating in the method section what statistical test was used to determine the significance (i.e., p=0.61).

-> As commented, we added the sentences as follows: Chi-square test was used for subgroup comparisons.

  1. Line 200. The reviewer recommends adding references to be consistent with “Several studies”.

-> As commented, we added the references.

[29] Imai Y, Nakane M, Kage K, Tsukahara S, Ishikawa E, Tsuruo T, Miki Y, Sugimoto Y. C421A polymorphism in the human breast cancer resistance protein gene is associated with low expression of Q141K protein and low-level drug resistance. Mol Cancer Ther. 2002;1(8):611-6.

[30] Mizuarai S, Aozasa N, Kotani H. Single nucleotide polymorphisms result in impaired membrane localization and reduced atpase activity in multidrug transporter ABCG2. Int J Cancer. 2004;109(2):238-46. doi: 10.1002/ijc.11669.

[31] Kondo C, Suzuki H, Itoda M, Ozawa S, Sawada J, Kobayashi D, Ieiri I, Mine K, Ohtsubo K, Sugiyama Y. Functional analysis of SNPs variants of BCRP/ABCG2. Pharm Res. 2004;21(10):1895-903. doi: 10.1023/b:pham.0000045245.21637.d4.

13 .Line 213-222. The reviewer recommends revising the whole paragraph. It is very difficult to comprehend.

-> As commented, we revised the sentences as follows: Subgroup analysis by ethnicity showed that, regardless of ethnicity, 421 A allele increased the rosuvastatin exposure by 50%. As the allele frequency of 421 A was higher in Asians than Caucasians (0.29 vs 0.15) [34], Asians may be more affected by this allele than Caucasians, thereby having increased exposure to rosuvastatin. In line with our results, the absolute oral bioavailability of rosuvastatin is higher in Asians than in Caucasians (29% vs 20%) [35]. The Food and Drug Administration (FDA) recommended reducing rosuvastatin dose in Asian patients, due to the increased toxicity of rosuvastatin in Asian patients [36].

  1. Line 223-225. The reviewer recommends revising the whole paragraph. It is very difficult to comprehend.

-> As commented we revised the sentences as follows: Another subgroup analysis by reported type of mean values showed that GM method and AM method showed similar results, supporting that the Higgins conversion equation [14] could provide reasonable estimates for pooling the results in the meta-analysis. Therefore, this equation can be an appropriate solution to include studies with both types of mean values in meta-analysis and derive strong evidence.